# Optimizing Hepatitis C Treatment Monitoring: Is Sustained Virologic Response at 4 Weeks Becoming the New Standard?

**DOI:** 10.3390/microorganisms12102050

**Published:** 2024-10-10

**Authors:** Ivana Milošević, Ana Filipović, Branko Beronja, Nikola Mitrović, Maja Ružić, Jelena Simić, Nataša Knežević, Maria Pete, Nevena Todorović, Nataša Nikolić

**Affiliations:** 1Clinic for Infectious and Tropical Diseases, University Clinical Center of Serbia, Bulevar Oslobođenja 16, 11000 Belgrade, Serbia; anafilipovic0211@gmail.com (A.F.); nikola.mitrovic@med.bg.ac.rs (N.M.); simicj093@gmail.com (J.S.); natasaknezevic995@gmail.com (N.K.); nevena.todorovic.1992@gmail.com (N.T.); 2Faculty of Medicine, University of Belgrade, Dr Subotica 8, 11000 Belgrade, Serbia; brankoberonj99@gmail.com; 3Faculty of Medicine, Clinic for Infectious Diseases, University Clinical Centre of Vojvodina, University of Novi Sad, 21000 Novi Sad, Serbia; maja.ruzic@mf.uns.ac.rs (M.R.); maria.pete@mf.uns.ac.rs (M.P.)

**Keywords:** chronic hepatitis C (CHC), sustained virological response 4 (SVR4), direct-acting antivirals (DAAs), predictive values

## Abstract

This study, conducted at two university-based infectious disease clinics, included 216 patients with chronic hepatitis C. The primary objective was to assess the positive and negative predictive values, sensitivity, and specificity of achieving a sustained virological response (SVR) at 4 weeks compared to 12 weeks post-therapy. The results demonstrated a maximum sensitivity of 100% for achieving SVR at 12 weeks after reaching SVR at 4 weeks for all analyzed genotypes, except for genotype 1b treated with EBR/GZR therapy, where the specificity was 75%. Additionally, younger age and less advanced liver fibrosis were identified as independent predictors of achieving a sustained virological response at both 4 and 12 weeks. The significant normalization of various biochemical parameters was observed after treatment, indicating an overall improvement in liver function. This study suggests that shortening the monitoring period to 4 weeks might be effective for younger patients without significant fibrosis, potentially reducing loss to follow-up, which is a critical issue in HCV treatment. These findings align with the “test and treat” approach. Further research is needed to confirm these findings and incorporate them into official guidelines, which could simplify and enhance the effectiveness of HCV treatment protocols, aiding global efforts to eliminate HCV as a public health issue by 2030.

## 1. Introduction

In the field of medicine, only a few diseases, such as hepatitis C virus (HCV) infection, have undergone a remarkable journey from their identification to their cure within the span of just a few decades. Untreated chronic HCV infection ultimately culminates in end-stage liver disease, including cirrhosis and hepatocellular carcinoma (HCC) [1,2,3]. The introduction of direct-acting antivirals (DAAs) with a stable virological response (SVR) rate exceeding 98%, minimal adverse effects, and short therapeutic protocols has revolutionized the paradigm of this disease [4,5,6,7,8,9]. SVR modifies the prognosis, resulting in improved clinical outcomes which prevent further progression of fibrosis and the development of liver cirrhosis and HCC [1]. This achievement led the World Health Organization (WHO) to set targets in 2016 for eliminating HCV infection as a public health issue by 2030 [10]. Target goals for HCV elimination encompass achieving a 90% decrease in both incidence and prevalence, providing treatment to 80% of individuals with chronic infection, reducing HCV-related deaths by 65%, and ensuring universal access to crucial prevention and treatment services.

Given that there is currently no vaccine, achieving the ultimate elimination of the virus necessitates the identification and treatment of undiagnosed cases. This involves the extensive screening of high-risk populations, including people who inject drugs (PWIDs), men who have sex with men (MSM), female sex workers, and prisoners [11,12]. In 2017, a mere 19% of the approximately 71 million people globally with CHC had received a diagnosis [13]. Furthermore, during the period of 2015 to 2016, merely 21% of those diagnosed had sought treatment [14]. Considering the plan for the elimination of hepatitis by 2030, WHO provided recommendations in 2022 for simplifying the path to the elimination of hepatitis C. In addition to other measures, this entails offering HCV testing and treatment in decentralized health facilities or community-based settings, such as primary care centers, harm-reduction sites, prisons, and HIV/ART clinics, among others [15]. Implementing these changes in middle-income countries is a complex task, involving substantial investments. Considering how the treatment process could be simplified without additional investments and systemic changes, we attempted to shorten the post-treatment follow-up period for treated patients. Antiviral treatment efficacy refers to sustained virological response assessed 12 weeks (SVR12) following cessation of treatment. In this pilot study, we analyzed the extent to which sustained virological response 4 weeks post-treatment (SVR4) corresponds to the SVR12 findings. The aim of this study was to analyze positive and negative predictive values, as well as the sensitivity and specificity of SVR4 in the attainment of SVR12. Additionally, the study aimed to investigate the predictive value of comorbidities, liver fibrosis stage, HCV genotype, and various laboratory parameters in achieving SVR4 and SVR12. Specifically, it sought to determine if there are differences in the factors that predict the achievement of SVR4 and SVR12.

## 2. Materials and Methods

This prospective study was conducted from August 2022 to December 2023 at two university clinics specializing in infectious diseases: the University Clinical Center of Serbia in Belgrade and the University Clinical Center of Vojvodina in Novi Sad. This study included 213 adult patients diagnosed with chronic hepatitis C (CHC).

The diagnosis of CHC was established in anti-HCV antibody-positive patients by the presence of HCV RNA in the blood for at least 6 months [16]. HCV RNA was determined in serum by a sensitive molecular method with a lower limit of detection ≤15 IU/mL (Cobas 4800 system, Roche Diagnostics, Sandhofer Strasse 116, 68305 Mannheim, Germany and Abbott m2000 RealTime System, Abbot GmbH, Max-Planck-Ring2, 65205 Wiesbaden, Germany).

Liver fibrosis was assessed by non-invasive methods—liver stiffness measurement (FibroScan^®^, Miami, FL, USA) or fibrosis-4 (FIB-4) index [17]. For a small number of patients, due to the inability to perform liver stiffness measurement or FIB-4 (excessive weight, high transaminases or combined liver-disease etiology), liver fibrosis was determined based on the pathohistological findings of liver tissue obtained through biopsy.

Patients were treated with DAAs according to EASL guidelines: elbasvir/grazoprevir (EBR/GZR), in the case of GT1b infection, and pangenotypic drugs glecaprevir/pibrentasvir (G/P) and sofosbuvir/velpatasvir +/− ribavirin (SOF/VEL +/− RBV) for other genotypes. HIV coinfection did not affect the choice of DAA therapy. [18]. Genotyping was performed using the cobas^®^ GT HCV genotyping test (Roche Diagnostics, Sandhofer Strasse 116, 68305 Mannheim, Germany) and the Abbott RealTime HCV Genotype II assay (Abbot GmbH, Max-Planck-Ring2, 65205 Wiesbaden, Germany) [15].

RBV was added to SOF/VEL in cases involving cirrhosis and genotype 3. The therapy with G/P was extended from 8 to 12 weeks for treatment-experienced patients and up to 16 weeks for treatment-experienced patients with cirrhosis and genotype 3. Patients with Child–Pugh B and C cirrhosis were treated exclusively with SOF/VEL regardless of genotype, as regimens containing protease inhibitors are contraindicated in these patients.

Both SVR12 and SVR24 endpoints have been approved by regulators in Europe and the United States, with a concordance rate exceeding 99%. In the DAA era, SVR12 has become the standard treatment endpoint, unlike the peg-IFN + RBV era, when SVR24 was relied upon [19,20]. Consequently, all included patients underwent PCR testing 12 weeks after treatment.

The positive predictive value (PPV) was calculated as the percentage of cases where both SVR4 and SVR12 were positive. The negative predictive value (NPV) was characterized as the percentage of patients who did not achieve SVR12 among those who failed to reach SVR4. Sensitivity is characterized as the proportion of patients with SVR4 among those who ultimately achieved SVR12. Specificity is defined as the proportion of patients without SVR4 among those who did not achieve SVR12. SVR4 was determined in all patients, regardless of the DAA protocol, associated comorbidities, fibrosis stage, and previous pegylated interferon and ribavirin (peg-IFN + RBV) therapy. NS3/4A protease-inhibitor- or NS5A inhibitor-experienced patients were not included in this study.

Participation in this study was voluntary. It included patients who were motivated to return for an additional examination and blood sampling 4 weeks after completing the therapy. All participants were informed about the study protocol, providing written informed consent, and the study was approved by the decision of the Ethics committee number 1264/14.

The data analysis was conducted using both descriptive and inferential statistical methods with IBM SPSS Statistics software, version 17.1 (IBM Corp., Armonk, NY, USA). A significance level of *p* < 0.05 was applied to determine statistical significance. Continuous variables were summarized as mean values and standard deviations, while categorical variables were represented by frequencies and percentages. The normality of continuous variable distributions was evaluated with the Kolmogorov–Smirnov test. The Wilcoxon signed-rank test, a non-parametric test for paired samples, was used to evaluate the significance of differences in laboratory parameters before and after treatment.

This study applied the Cox proportional hazards model to investigate factors associated with achieving SVR4 and SVR12. Initially, all variables underwent univariate analysis. Subsequently, variables demonstrating statistical significance (*p* < 0.005) in the univariate model proceeded to the multivariate analysis. To ensure model robustness and prevent overfitting, variables were categorized into three distinct models: in the first model, comorbidities, coinfections and lifestyle habits were included as predictors; the second model incorporated liver fibrosis stage; lastly, the third model focused on laboratory parameters. Each of these models was adjusted for sex, age and the degree of liver fibrosis to account for potential confounding factors and enhance the reliability of the findings.

## 3. Results

In this study, conducted in two university infectious-diseases clinics, 216 participants were included, with the majority being male (125, 57.9%). The average age of participants was 51.21 ± 13.74 years, with no significant age difference between genders (*p* = 0.299). Of the 216 subjects, 81 (37.5%) were PWID. Hypertension was the most common comorbidity, present in 50 participants (23.1%). Additionally, 15 participants (6.9%) were coinfected with HIV, and 4 participants (1.8%) had HBV infection. The distribution of other comorbidities is detailed in Table 1.

The predominant genotype was GT1 (with subtype 1a being the most common), confirmed in 79 patients (41.8%), and GT3, found in 74 patients (39.1%), followed by genotype 4, diagnosed in 20 patients (9.6%). All of them were treated with pangenotypic regimens: G/P in 121 patients (56.0%) and SOF/VEL +/− RBV in 68 patients (31.5%). The degree of liver fibrosis was assessed using various methods, with liver stiffness measurement being the most common technique employed in 108 (50.0%) patients. A significant number of patients, 102 (47.2%), were classified in the F0/F1 fibrosis category. Additionally, 44 patients (20.3%) developed liver cirrhosis before DAAs introduction, and 8 patients (3.7%) were diagnosed with HCC. A more detailed overview of the therapeutic and diagnostic modalities used in the study is provided in Table 2. The laboratory parameters measured immediately before the initiation of treatment and 4 weeks after its completion are presented in Appendix A. There is statistically significant normalization observed in serum levels of total bilirubin, direct bilirubin, aspartate aminotransferase, alanine aminotransferase, gamma-glutamyl transferase, alkaline phosphatase, and alpha-fetoprotein, all with *p*-values < 0.001. Additionally, normalization is noted in albumin levels (*p* = 0.018) and fibrinogen levels (*p* = 0.008), as is a significant increase in platelet count (*p* = 0.026).

### 3.1. The PPV, NPV, Sensitivity and Specificity of SVR4 for Achieving SVR12

All patients treated with pangenotypic drugs, regardless of genotype, achieved SVR4 and SVR12 at the same rate. It was observed that SVR4 and SVR12 were not achieved in (16.7%) patients treated with SOF/VEL with genotype 1b, as well as in 1 (3.4%) patient treated with SOF/VEL with genotype 1a. This corresponds to an overall SVR rate of 98.9%, which is identical at both 4 and 12 weeks. Sensitivity is defined as the proportion of positive outcomes that are correctly identified as such. In the studied sample, a maximum sensitivity of 100% was observed for achieving SVR12 after reaching SVR4 across all analyzed genotypes. Specificity is defined as the proportion of negative outcomes that are correctly identified as such. In the studied sample, the average specificity for achieving SVR12 after reaching SVR4 was 80%; but in the subgroup of patients treated with pangenotypic DAAs, specificity was 100%. The negative predictive value for the entire sample was maximized at 100%. Similarly, a maximum positive predictive value of 100% was recorded (Figure 1).

### 3.2. Analysis of Predictors of SVR4 and SVR12

The analysis across all three Cox multivariate regression models identified three variables that were independently associated with higher rates of achieving SVR4 and SVR12: younger age and less advanced fibrosis stage. Other analyzed variables were not independently associated with the rates of achieving SVR4 and SVR12. A comprehensive analysis of the predictors is detailed in Table 3 and Appendix A.

## 4. Discussion

There are numerous examples of paradigm shifts in human medicine, especially in infectious diseases, due to the discovery of vaccines and antimicrobial therapy. However, one of the major advancements in medical science was the introduction of DAAs in the treatment of CHC. Within a few decades of the discovery of HCV in 1989, the introduction of DAAs with high activity and low toxicity has made it possible to set ambitious goals, such as eliminating chronic hepatitis C as a public health issue by 2030. This necessitates diagnosing 90% of individuals with CHC, treating 80% of those diagnosed with the intent to cure, and implementing additional measures to reduce the incidence of HCV in high-risk populations [21,22]. However, the realization of these goals in practice requires much work and the overcoming of numerous global gaps. Considering that HCV is mostly an asymptomatic disease, it is not surprising that only 50% of those infected are aware of their infection in highly developed countries such as the USA, and probably significantly less (around 20%) in low- and middle-income countries [23,24,25,26]. Additionally, the COVID-19 pandemic has significantly hindered the achievement of WHO goals by disrupting all stages of the viral hepatitis care process, affecting various centers similarly.

Achieving the WHO goals requires the engagement of different stakeholders, beginning with the efforts of practitioners in their routine practice. Eliminating HCV as a public health issue should be viewed as a puzzle where even the smallest piece is crucial for completing the big picture, turning a public health crisis into a successfully resolved problem. From the perspective of practicing physicians, a significant issue can be loss to follow-up (LTFU) at any step of the HCV cascade of care [27]. Hence, we conducted this study to assess early SVR control at 4 weeks and compare it with SVR12. This approach would shorten the confirmation period for SVR and thereby decrease LTFU among patients who have completed DAA therapy. A previous study even showed that a lack of post-treatment follow-up had no impact on SVR rates [28].

Intention-to-treat (ITT) SVR rates indicate the proportion of patients achieving SVR among those who started DAA therapy. In studies involving mixed populations, ITT SVR varies widely from 22% to 98% (median 83%) [29,30,31,32,33]. It is interesting to analyze ITT SVR rates among PWID and HIV/HCV-coinfected populations, where ITT SVR ranges from 80% to 92% (median 85%) and 80% to 96% (median 91%), respectively [27,34,35,36,37,38,39,40].

In order to eliminate HCV as a public health issue, it is particularly important to focus on inmates, who are a core population in terms of generating new cases of HCV infection. It is projected that over 10.2 million people globally, including both sentenced individuals and pre-trial detainees, are held in correctional facilities [41]. PWID are significantly over-represented in prisons, often making up 50% of the inmate population. As a result, the transmission of HCV and other blood-borne infections poses a serious problem in many prison systems [42]. The mass treatment of inmates, with the reduction in SVR determination time to 4 weeks, could be an effective way to address the issue of HCV infection in this population. In addition, the literature mentions many other risk factors for LTFU. Younger age (≤45 years), hospital-based treatment, a history of homelessness, mental illness, and insurance status were among the most frequently cited factors associated with LTFU [27].

SVR24 has been widely acknowledged as a reliable marker of cure and has served as a surrogate endpoint linked to decreased risks of HCC, liver decompensation, and liver-related and overall mortality [43,44,45,46].

Chen et al., discovered that SVR12 and SVR24 measurements demonstrated consistency across a significant cohort of individuals with CHC participating in clinical trials with varied treatment protocols and durations [47]. They concluded that SVR12 could be established as a suitable primary endpoint for regulatory approval, while SVR4 was potentially considered for guiding dosing and treatment strategies during trials [47].

The introduction of SOF (a NS5B polymerase inhibitor) led the authors to consider shortening the period required to define SVR already in the era of therapy protocols based on SOF [48]. As the introduction of SOF has enabled the shortening of therapy, SVR4, SVR12 and SVR24 were analyzed in patients treated with SOF-plus-RBV with or without peg-IFN, depending on the HCV genotype. In patients treated with the mentioned protocol, the PPV of SVR12 for SVR24 was 99.7%, and the NPV was 100%. These high values indicate a very reliable prediction treatment outcome at SVR24 based on achieving SVR12 [48]. Even more intriguingly, during this therapeutic protocol—regarded as inferior to current IFN-free protocols—the PPV of SVR4 for SVR12 was 98.0%, coupled with a flawless NPV of 100% [24]. This suggests a high reliability in predicting treatment success early on.

Given that DAA therapies typically extend over 8–12 weeks, we aimed to evaluate the possibility of further shortening the time to SVR. This approach could align with the “test and treat” concept promoted by EASL and many authors [49,50]. If not applicable to every individual within the CHC cohort, SVR4 could at least be considered for implementation in patients without significant fibrosis. Gane and colleagues investigated the possibility of implementing SVR4 in patients treated with G/P [51]. That was the starting point for us to explore the possibility of implementing SVR4 in patients treated with other therapeutic options. Analyzing the therapeutic options recommended by EASL and AASLD, we identified the first limitation for the universal application of SVR4 as the end of follow-up. There was no statistically significant difference in achieving SVR4 and SVR12 among patients with genotypes 1a, 1b, 2, 3 and 4 treated with pangenotypic options. A significant discrepancy in achieving SVR12 compared to SVR4 (90% vs. 93.3%) was observed in the subgroup of patients treated with EBR/GZR. This leads to the conclusion that SVR4 could be used as a new treatment endpoint for patients treated with G/P or SOF/VEL. The discouraging result in sensitivity observed in the subgroup treated with EBR/GZR should be interpreted with caution due to the small number of patients treated with this therapeutic option. There are further arguments in favor of applying the concept of SVR4 to pangenotypic treatments. Both G/P and SOF/VEL have demonstrated a maximal sensitivity and specificity (100%) of SVR4 for predicting a favorable SVR12. In addition, the PPV and NPV of SVR4 were also maximally 100% for these two therapeutic options. These results align with published data on patients treated with SOF-based regimens, where the PPV of SVR4 for achieving SVR12 was over 98% and the NPV was 100% [48]. Gane et al., found similar rates for PPV and NPV in patients treated with G/P, at 99.8% and 100%, respectively [51].

It is particularly interesting to define the characteristics of patients who are suitable for the application of shortened post-treatment monitoring. The results of this study showed that younger patients and those with less advanced disease are ideal candidates for a shorter period of post-treatment follow-up, using SVR4 as an end-treatment goal. Shorter post-treatment monitoring for less advanced disease is part of EASL recommendations, as continued follow-up after treatment is advised for individuals with cirrhosis and F3 fibrosis due to HCC screening [43]. HIV positivity does not influence the selection of therapy options or treatment outcomes according to EASL guidelines, and, as reported in this study, it also does not impact the achievement of SVR4. [15]. The results of the study also showed a significant improvement in biochemical parameters after HCV infection is cured. The effects of HCV cure can be observed very early after achieving viral clearance. In this study, significant improvement in platelet numbers was already visible 4 weeks after the end of therapy. Furthermore, improvements in bilirubin levels, transaminases, alkaline phosphatase and gamma-glutamyl transpeptidase have been achieved due to the cessation of chronic infection. Lastly, improvement in liver synthetic function has been evidenced by the increase in albumin and fibrinogen levels. These results are consistent with the relevant literature and indicate an unquestionable benefit of HCV cure [52]. However, it should be emphasized that further follow-up is necessary even after achieving SVR if there is a lack of biochemical response.

It is important to note the limitations of this study. Firstly, although the results are consistent with available data, they would be more convincing if the sample size had been larger. This is particularly relevant to the conclusions related to EBR/GZR. Specifically, the smallest number of patients in the study were treated with this therapeutic option, and it is important to note that the SVR12 in these patients is lower than what the authors’ experience with this medication would suggest. The authors recognize this as a limitation of the study; however, they believe it still provides valuable insights and can stimulate further research in this area. It is also important to consider that the treatment of patients was confined to university centers due to funding constraints. While this is not an isolated occurrence, it significantly contrasts with practices in some other countries. Nevertheless, despite the aforementioned biases, the authors believe that the results support the use of SVR4 for younger individuals without significant fibrosis, who have an adequate biochemical response to pangenotypic treatment [53]. Further studies to support these and similar findings are, of course, necessary. All of this would be just one step in simplifying the HCV cascade in an effort to reach the global goal of eliminating CHC as a public health challenge.

## 5. Conclusions

Eliminating HCV infection as a public health problem remains a challenge, despite the availability of highly effective and well-tolerated therapies. Many different obstacles persist, despite global efforts to achieve this goal. Attempting to shorten the monitoring period after completing therapy could, at least for some patients, simplify this process. Therefore, SVR4 as an end-treatment goal could be used for younger individuals without significant fibrosis who demonstrate a good biochemical response to pangenotypic regimens. This could be part of a “test and treat” approach for vulnerable populations (PWID, inmates, etc.). Positive results regarding the use of SVR4 from this study and published data require further research and incorporation into official guidelines before they can become part of routine practice.

## Figures and Tables

**Figure 1 microorganisms-12-02050-f001:**
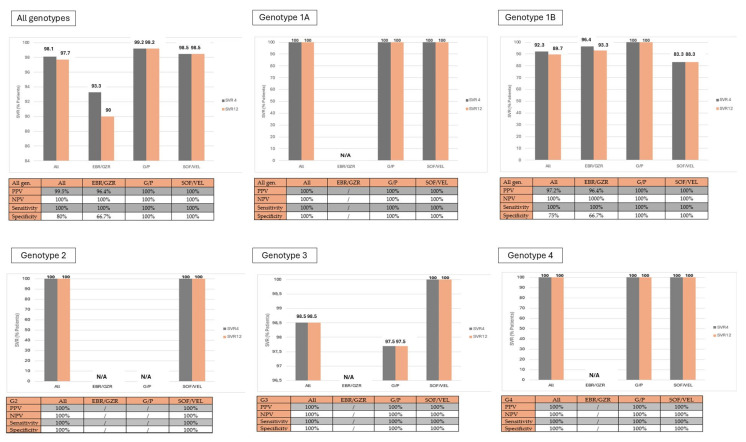
Graphical representation of achieving SVR4 and SVR12; sensitivity, specificity, PPV and NPV of SVR4 required for achieving SVR12 by genotypes according to the treatment modality; G/P—glecaprevir/pibrentasvir; SOF/VEL—sofosbuvir/velpatasvir +/− ribavirin.

**Table 1 microorganisms-12-02050-t001:** Overview of demographic characteristics, comorbidities, coinfections and substance use habits; CTD—connective tissue disease.

Variable		N = 216
Age (mean ± SD)		51.21 ± 13.74
Sex, *n* (%)	Male	125 (57.9)
	Female	91 (42.1)
Chronic diseases, *n* (%)	Hypertension	50 (23.1)
	Other CV diseases	15 (6.9)
	Diabetes mellitus	25 (11.6)
	Respiratory diseases	10 (4.6)
	Chronic kidney failure	8 (3.7)
	Dialysis	5 (2.3)
	Malignant diseases	13 (6.0)
	CTD	4 (1.8)
	Cryoglobulinemia	1 (0.5)
	Epilepsy	6 (2.7)
	Hypothyroidism/hyperthyroidism	13 (6.0)
	Mood disorders	14 (6.5)
	Psychoses	8 (3.7)
Coinfections, *n* (%)	HIV	15 (6.4)
	Antiretroviral therapy	13 (6.0)
	HBV	4 (1.8)
	Resolved HBV	30 (13.9)
Substance use disorders, *n* (%)	Chronic alcoholics	21 (9.7)
	Intravenous drug users	81 (37.5)

**Table 2 microorganisms-12-02050-t002:** Evaluation of diagnostic and therapeutic characteristics and modalities in patient management; md—mediana; IQR—interquartile range; HCC—hepatocellular carcinoma; RFA—radiofrequency ablation; TACE—trans-arterial chemoembolization.

Variable		N = 216
HCV RNA quantitative testing, median (IQR)	523,500.0 (18,310.0–2,235,400.0)	
Sustained virologic response, *n* (%)	at post-treatment Week 4	212 (98.1)
	at post-treatment Week 12	211 (97.7)
Hepatitis C genotype, *n* (%)	1a	79 (36.5)
	1b	39 (18.1)
	2	5 (2.3)
	3	73 (33.8)
	4	20 (9.3)
Antiviral therapy, *n* (%)	Glecaprevir/pibrentasvir	118 (54.6)
	Elbasvir/grazoprevir	30 (13.9)
	Sofosbuvir/velpatasvir +/− ribavirin	68 (36.0)
Method of liver fibrosis assessment, *n* (%)	Fibrosis-4 (FIB-4) index	61 (28.2)
	Liver stiffness measurement	127 (58.8)
	Liver biopsy	28 (13.0)
Fibrosis stage, *n* (%)	F0/1	113 (52.3)
	F2	34 (15.8)
	F3	16 (7.4)
	F4	53 (24,5)
Liver stiffness measurement	F0/1, *n* (%)	74 (34.3)
	F2, *n* (%)	21 (9.7)
	F3, *n* (%)	6 (2.8)
	F4, *n* (%)	17 (7.9)
	Fibrosis stage kPa, md (IQR)	6.5 (5.0–9.2)
	Steatosis S0, *n* (%)	69 (31.9)
	Steatosis S1, *n* (%)	26 (12.0)
	Steatosis S2/3, *n* (%)	19 (8.8)
	Degree of steatosis dB/m, md (IQR)	215.6 (190.1–242.5)
Fibrosis-4 index, *n* (%)	F0/1	34 (15.8)
	F2/3	10 (4.6)
	F3/4	26 (12.0)
Liver cirrhosis and Child–Pugh classification	Total cirrhotic patients	50 (23.1)
	Class A	39 (18.0)
	Class B	10 (4.6)
	Class C	1 (0.5)
Complications of liver cirrhosis	Ascites, *n* (%)	12 (5.5)
	Hepatic encephalopathy, *n* (%)	5 (2.3)
	Portal hypertension, *n* (%)	22 (10.1)
	Esophageal varices, *n* (%)	20 (9.3)
Hepatocellular carcinoma and treatment modalities, *n* (%)	Total HCC patients	10 (4.2)
	Surgical resection	5 (2.3)
	RFA/TACE	3 (1.4)
	Palliative care	2 (0.9)

**Table 3 microorganisms-12-02050-t003:** Results of the Cox proportional hazard models: factors associated with SVR4 and SVR12; HR-hazard ratio; CI-confidence interval; CTD—connective tissue disease; PWID—people who inject drugs.

Model 1	SVR4						SVR12					
Variable	Univariante			Multivariate			Univariante			Multivariate		
	HR	95% CI	*p*	HR	95% CI	*p*	HR	95% CI	*p*	HR	95% CI	*p*
Sex	0.919	0.61–1.23	0.401				0.836	0.48–1.21	0.332			
Ages	0.951	0.91–0.99	0.003	0.991	0.90–1.04	0.194	0.921	0.89–0.97	0.001	1.014	0.74–1.34	0.849
Stage of liver fibrosis	0.801	0.69–0.97	0.001	0.855	0.68–0.96	0.019	0.884	0.75–0.96	0.002	0.964	0.95–0.99	0.011
Hypertension	0.738	0.45–1.17	0.154				0.742	0.44–1.28	0.151			
Other CV diseases	0.575	0.35–1.36	0.224				0.647	0.32–1.27	0.298			
Diabetes mellitus	0.861	0.51–1.28	0.234				0.952	0.62–1.85	0.936			
Respiratory disease	0.942	0.35–1.43	0.207				0.636	0.41–1.36	0.487			
Chronic kidney failure	0.671	0.42–1.73	0.651				0.721	0.49–1.21	0.645			
Dialysis	0.669	0.26–1.97	0.501				0.984	0.30–2.08	0.670			
Malignant diseases	0.792	0.50–1.37	0.436				0.169	0.40–1.23	0.569			
CTD	0.357	0.14–1.18	0.969				0.401	0.12–1.24	0.290			
Cryoglobulinemia	0.895	0.01–5.25	0.326				0.811	0.11–5.87	0.598			
Epilepsy	1.198	0.36–2.26	0.424				1.236	0.74–2.98	0.657			
Hypo/hyperthyroidism	0.893	0.48–1.43	0.845				1.062	0.97–1.83	0.941			
Mood disorders	1.158	0.75–1.49	0.360				1.148	0.67–1.58	0.290			
Psychoses	0.821	0.40–1.22	0.460				0.911	0.54–1.62	0.215			
HIV	0.548	0.40–0.95	0.004	0.758	0.50–1.17	0.187	0.277	0.53–0.99	0.004	0.851	0.60–1.17	0.187
Antiretroviral therapy	1.160	0.62–1.32	0.120				1.760	0.85–2.59	0.120			
HBV	0.974	0.78–1.25	0.521				0.622	0.21–2.58	0.240			
Resolved HBV	0.929	0.49–1.43	0.309				0.861	0.41–1.74	0.680			
Chronic alcoholics	0.479	0.20–1.18	0.525				0.388	0.06–1.25	0.490			
PWID	1.870	1.01–1.99	0.002	2.912	0.93–4.80	0.068	1.481	1.10–1.96	0.001	1.297	0.85–1.90	0.174
**Model 2**	**SVR4**						**SVR12**					
**Variable**	**Univariante**			**Multivariate**			**Univariante**			**Multivariate**		
	**HR**	**95% CI**	** *p* **	**HR**	**95% CI**	** *p* **	**HR**	**95% CI**	** *p* **	**HR**	**95% CI**	** *p* **
Sex	0.919	0.61–1.23	0.401				0.836	0.48–1.21	0.332			
Ages	0.951	0.91–0.99	0.003	0.878	0.88–0.95	0.001	0.921	0.89–0.97	0.001	1.089	0.41–1.29	0.950
Stage of liver fibrosis	0.801	0.69–0.97	0.001	0.914	0.80–0.97	0.003	0.884	0.75–0.96	0.002	0.870	0.74–0.98	0.007
HCV RNA quantitative	1.970	1.13–201	0.278				1.118	1.03–1.15	0.281			
Hepatitis C genotype	1.541	0.79–2.17	0.731				0.527	0.92–1.12	0.754			
Fibrosis-4 index	0.881	0.44–0.98	0.034	0.811	0.71–1.17	0.245	0.745	0.57–1.08	0.041	0.895	0.54–1.59	0.256
Child–Pugh class	1.001	0.30–1.97	0.541				1.035	0.57–1.98	0.857			
Degree of fibrosis kPa	2.095	0.91–4.64	0.098	0.979	0.68–1.84	0.167	0.917	0.84–0.98	0.036	0.887	0.75–1.48	0.398
Degree of steatosis bD/m	1.045	0.94–1.42	0.541				1.004	0.99–1.09	0.504			
Advanced complications	0.687	0.52–0.77	0.012	0.684	0.39–1.74	0.241	0.428	0.32–0.58	0.020	0.642	0.41–1.18	0.210
Hepatocellular carcinoma	0.556	0.31–1.89	0.314				0.679	0.35–1.29	0.128			

## Data Availability

Data is unavailable due to privacy or ethical restrictions.

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
