# Peer review of "Optimizing Hepatitis C Treatment Monitoring: Is Sustained Virologic Response at 4 Weeks Becoming the New Standard?"

_microorganisms, 2024, doi:10.3390/microorganisms12102050_

Round 1
Reviewer 1 Report (Previous Reviewer 2)
Comments and Suggestions for Authors
The authors analyzed the PPV and NPV of SV4 (testing for HCV RNA at week 4) compared to SV12, after DAA treatment. Some concerns hamper the acceptance of this manuscript.
1. This manuscript is a resubmission. The number of samples changed, 200 in the original version vs. 189 in this one, although in the first submission there was an additional treatment with 30 patients (Elbasvir/grazoprevir). With this particular treatment, there was a discordance between SVR4 and SVR12. This should be explained, and even if the EBR/GZR are not included in the new version, this discordance should be mentioned,
2. The number of patients analyzed is still low and this should be stressed in the Discussion.
3. In addition, the Discusion is somehow long and many parts do not refer to the results of this study.
Author Response
The authors analyzed the PPV and NPV of SV4 (testing for HCV RNA at week 4) compared to SV12, after DAA treatment. Some concerns hamper the acceptance of this manuscript.
This manuscript is a resubmission. The number of samples changed, 200 in the original version vs. 189 in this one, although in the first submission there was an additional treatment with 30 patients (Elbasvir/grazoprevir). With this particular treatment, there was a discordance between SVR4 and SVR12. This should be explained, and even if the EBR/GZR are not included in the new version, this discordance should be mentioned,
Comment 1. We thank the reviewer 1 for their critical reading and suggestions. The first version of the manuscript included patients treated with the EBR/GZR option, with the intention of analyzing all three therapeutic options that are part of the HCV treatment guidelines (EBR/GZR, G/P, SOF/VEL). The authors themselves noted in the first version of the manuscript that the number of patients treated with the EBR/GZR option was small, and that the SVR achievement results with this drug among the patients included in the study do not align with global data or the authors' own experience. The authors have treated a significantly larger number of patients with the EBR/GZR regimen and achieved excellent results, with an SVR rate of around 98%. In the small sample of patients where SVR4 was assessed, a lower SVR rate was observed. The authors can only speculate about the reasons for this—potentially inadequate compliance, reinfection, or other factors. This was precisely the reason for the discordance between SVR4 and SVR12 in the group treated with EBR/GZR.
Since the potential strategy of determining SVR4 instead of SVR12, aimed at simplifying the HCV care cascade, is still an experimental approach adopted by a few authors worldwide (from Taiwan, New Zealand, and now from Serbia as well), the authors believed it was important to present the results with EBR/GZR despite the small sample size and the poorer treatment outcomes compared to both the authors' own experience and global findings. The authors emphasized this in the initial version of the manuscript, noting that while the results with EBR/GZR are not representative, they should still be presented and serve as a starting point for further research by both the authors and others. It was not possible to increase the sample size or improve the results, as the number of patients treated with EBR/GZR included in the study was small.
During the review process, the authors received results for an additional group of patients treated with a pangenotypic option, but not EBR/GZR (only one more patient, who would not significantly impact the results). This was the sole reason the authors decided to exclude this subgroup from their study, as the initial reviewers did not consider it adequate, based on the explanation provided by the authors.
However, due to Your request in revised version of resubmitted manuscript, the authors were more than happy to provide the EBR/GZR results afterall. So, we have provided detailed data in an Excel chart for all included participants, made new statistical analysis and all new results, regarding treatment with all treatment options (G/P, SOF/VEL, EBR/GZR) are given in this revised version of the resubmitted manuscript.
Since the data changed, we are sending the new dataset as an attachment in the mail to the editor, Sylvia Shi, because we were not sure how to provide new dataset.
We must note that the study from New Zealand, which analyzes SVR4, includes only patients treated with one of the pangenotypic options (G/P, not SOF/VEL). The authors humbly believe that their research provides additional insights into the potential role of SVR4, as it investigates both pangenotypic options and EBR/GZR.
The number of patients analyzed is still low and this should be stressed in the Discussion.
Comment 2. Regarding the sample size, the authors themselves emphasize this in the discussion as one of the limitations of the study. This is due to the fact that the authors reside in a small nation with an HCV infection prevalence of approximately 0,55 - 1%. When comparing the sample size from the authors' country to the sample from the SVR4 study in Taiwan, it can be observed that the samples are proportional to the populations of these two countries. Considering the sample size, prior to writing the manuscript, statistical analysis indicated that the sample is sufficiently large to provide statistical power for drawing conclusions. The authors do not think that their results are decisive for future practice, but they believe that they can contribute to ongoing efforts in the control of HCV.
In addition, the Discussion is somehow long, and many parts do not refer to the results of this study.
Comment 3. The authors have shortened the discussion as advised by the reviewer 1. We would like to note that there are only a few studies on SVR4, which was one of the reasons for conducting this study. All relevant studies published to date are mentioned in the discussion. Given the brief history of the DAA era, the authors address the transition from SVR24 to SVR4 during the shift from IFN-based therapy to DAA therapy, further highlighting the potential role of SVR4 by referencing published studies. The authors also pointed out in the two core HCV populations: inmates and PWID. Young newly infected individuals primarily come from these two subgroups, and it was emphasized in the discussion that determining SVR4 would be most meaningful within these two groups. The discussion has been shortened as required, and the authors hope it will now meet the reviewer 1's criteria.
Reviewer 2 Report (Previous Reviewer 1)
Comments and Suggestions for Authors
The manuscript “Optimizing hepatitis C treatment monitoring: is sustained virologic response at 4 weeks becoming the new standard?” submitted to the journal Microorganisms is deal with the assessment the positive and negative predictive values, sensitivity, and specificity of achieving sustained virological response at 4 weeks compared to 12 weeks after therapy. The topic discussed is very important for the improving hepatitis C treatment monitoring, and to shorten the monitoring period after completing therapy and could simplify this process which is especially relevant in low-income countries.
The authors took the recommendations into account and revised the manuscript. I have only one comment: Figure 1 is not legible, it would be better to enlarge it.
Author Response
The manuscript “Optimizing hepatitis C treatment monitoring: is sustained virologic response at 4 weeks becoming the new standard?” submitted to the journal Microorganisms is deal with the assessment the positive and negative predictive values, sensitivity, and specificity of achieving sustained virological response at 4 weeks compared to 12 weeks after therapy. The topic discussed is very important for the improving hepatitis C treatment monitoring, and to shorten the monitoring period after completing therapy and could simplify this process which is especially relevant in low-income countries.
The authors took the recommendations into account and revised the manuscript. I have only one comment: Figure 1 is not legible, it would be better to enlarge it.
Comments 1. The authors are deeply grateful for Reviewer 2’s feedback and have made the requested corrections.
Round 2
Reviewer 1 Report (Previous Reviewer 2)
Comments and Suggestions for Authors
The authors addressed satisfactorely the concerns.
This manuscript is a resubmission of an earlier submission. The following is a list of the peer review reports and author responses from that submission.
Round 1
Reviewer 1 Report
Comments and Suggestions for Authors
The manuscript “Optimizing hepatitis C treatment monitoring: is SVR4 the new SVR12?” submitted to the journal Microorganisms is deal with the assessment the positive and negative predictive values, sensitivity, and specificity of achieving sustained virological response at 4 weeks compared to 12 weeks after therapy. The topic discussed is very important for the improving hepatitis C treatment monitoring, and to shorten the monitoring period after completing therapy and could simplify this process which is especially relevant in low-income countries.
I would like to make a few comments:
1. It is advisable to change the title and not use abbreviations.
2. It is better not to use abbreviations in the abstract.
3. In the Materials and Methods section, it is advisable to provide treatment plans for patients.
4. In the results section, it is advisable to discuss the change in the cellular composition of the blood after therapy, depicted in Supplementary table.
Author Response
Responses to reviewer 1
Dear Reviewer,
We would like to express our sincere gratitude for your thorough and constructive review of our manuscript. Your insightful feedback and detailed comments have been invaluable in enhancing the quality of our work.
We would like to inform the reviewers that the revised version of the manuscript includes an additional 6 patients (3 with GT1b, 2 with GT1a, and 1 with GT3) due to a subsequent evaluation of the data. The final number of included participants is 206. All statistical data have been appropriately revised. We hope that this addition further enhances the informativeness of our study.
Comments 1. It is advisable to change the title and not use abbreviations.
Response 1. Dear Reviewer, the authors have considered your suggestion and have changed the title to: “Optimizing Hepatitis C Treatment Monitoring: Is Sustained Virologic Response at 4 Weeks Becoming the New Standard?”
Comments 2. It is better not to use abbreviations in the abstract.
Response 2. We have also removed abbreviations from the abstract.
Comments 3. In the Materials and Methods section, it is advisable to provide treatment plans for patients.
Response 3. In the first version, the authors refer to EASL guidelines. The medications used to treat patients were listed, with a note that two therapeutic options (SOF/VEL and G/P) are pangenotypic, while elbasvir/grazoprevir is used specifically for genotype 1b. In the revised version of the manuscript, the following has also been added: “HCV-infected patients without cirrhosis and those with Child-Pugh A cirrhosis, including both treatment-naïve and treatment-experienced patients (previously treated with peg-IFN+RBV), regardless of their HIV status, were treated with the DAAs. RBV was added to SOF/VEL in cases involving cirrhosis and genotype 3. The therapy with G/P was extended from 8 to 12 weeks for treatment-experienced patients and up to 16 weeks for treatment-experienced patients with cirrhosis and genotype 3. Patients with Child-Pugh B and C cirrhosis were treated exclusively with SOF/VEL regardless of genotype, as regimens containing protease inhibitors are contraindicated in these patients.” (page 3, and lines 90-97)
Comments 4. In the results section, it is advisable to discuss the change in the cellular composition of the blood after therapy, depicted in Supplementary table.
Response 4. The authors have already presented changes in the cellular composition of the blood after therapy (page number 5, lines 156-161), and they have improved the discussion in the revised manuscript with reference to these results by adding the following: “The cure of HCV infection prevents further progression of fibrosis and the development of cirrhosis and HCC. The effects of HCV cure can be observed very early after achieving viral clearance. In this study, significant improvement in platelet numbers was already visible 4 weeks after the end of therapy. Furthermore, improvements in bilirubin levels, transaminases, alkaline phosphatase, and gamma-glutamyl transpeptidase have been demonstrated, as a result of the cessation of chronic damage induced by the virus during chronic infection. Finally, improvement in liver synthetic function has been demonstrated through the enhancement of albumin and fibrinogen levels. These results are consistent with the relevant literature and indicate an unquestionable benefit of HCV cure. (page 10, lines 296-305).
This change in the discussion required a new reference, which has also been added: “Ferreira, J.; Bicho, M.; Serejo, F. Effects of HCV Clearance with Direct-Acting Antivirals (DAAs) on Liver Stiffness, Liver Fibrosis Stage and Metabolic/Cellular Parameters. Vir. 2024, 16, 371. https://doi.org/10.3390/v16030371"
Responses to reviewer 1
Dear Reviewer,
We would like to express our sincere gratitude for your thorough and constructive review of our manuscript. Your insightful feedback and detailed comments have been invaluable in enhancing the quality of our work.
We would like to inform the reviewers that the revised version of the manuscript includes an additional 6 patients (3 with GT1b, 2 with GT1a, and 1 with GT3) due to a subsequent evaluation of the data. The final number of included participants is 206. All statistical data have been appropriately revised. We hope that this addition further enhances the informativeness of our study.
Comments 1. It is advisable to change the title and not use abbreviations.
Response 1. Dear Reviewer, the authors have considered your suggestion and have changed the title to: “Optimizing Hepatitis C Treatment Monitoring: Is Sustained Virologic Response at 4 Weeks Becoming the New Standard?”
Comments 2. It is better not to use abbreviations in the abstract.
Response 2. We have also removed abbreviations from the abstract.
Comments 3. In the Materials and Methods section, it is advisable to provide treatment plans for patients.
Response 3. In the first version, the authors refer to EASL guidelines. The medications used to treat patients were listed, with a note that two therapeutic options (SOF/VEL and G/P) are pangenotypic, while elbasvir/grazoprevir is used specifically for genotype 1b. In the revised version of the manuscript, the following has also been added: “HCV-infected patients without cirrhosis and those with Child-Pugh A cirrhosis, including both treatment-naïve and treatment-experienced patients (previously treated with peg-IFN+RBV), regardless of their HIV status, were treated with the DAAs. RBV was added to SOF/VEL in cases involving cirrhosis and genotype 3. The therapy with G/P was extended from 8 to 12 weeks for treatment-experienced patients and up to 16 weeks for treatment-experienced patients with cirrhosis and genotype 3. Patients with Child-Pugh B and C cirrhosis were treated exclusively with SOF/VEL regardless of genotype, as regimens containing protease inhibitors are contraindicated in these patients.” (page 3, and lines 90-97)
Comments 4. In the results section, it is advisable to discuss the change in the cellular composition of the blood after therapy, depicted in Supplementary table.
Response 4. The authors have already presented changes in the cellular composition of the blood after therapy (page number 5, lines 156-161), and they have improved the discussion in the revised manuscript with reference to these results by adding the following: “The cure of HCV infection prevents further progression of fibrosis and the development of cirrhosis and HCC. The effects of HCV cure can be observed very early after achieving viral clearance. In this study, significant improvement in platelet numbers was already visible 4 weeks after the end of therapy. Furthermore, improvements in bilirubin levels, transaminases, alkaline phosphatase, and gamma-glutamyl transpeptidase have been demonstrated, as a result of the cessation of chronic damage induced by the virus during chronic infection. Finally, improvement in liver synthetic function has been demonstrated through the enhancement of albumin and fibrinogen levels. These results are consistent with the relevant literature and indicate an unquestionable benefit of HCV cure. (page 10, lines 296-305).
This change in the discussion required a new reference, which has also been added: “Ferreira, J.; Bicho, M.; Serejo, F. Effects of HCV Clearance with Direct-Acting Antivirals (DAAs) on Liver Stiffness, Liver Fibrosis Stage and Metabolic/Cellular Parameters. Vir. 2024, 16, 371. https://doi.org/10.3390/v16030371"
Reviewer 2 Report
Comments and Suggestions for Authors
The authors analyzed the PPV and NPV of SV4 (testing for HCV RNA at week 4) compared to SV12, after DAA treatment. Some concerns hamper the acceptance of this manuscript.
1. Materials and Methods, page 2, lines 93-98. Some sentences are confusing throughout the manuscript, particularly the using of sensitivity and specificity for SV4 and SV12. For example: ¨ Negative predictive value (NPV) was characterized as the percentage of patients lacking SVR12 among those who did not attain SVR4¨.
2. Figure 1: Genotype 1A. The number 0 is not correct. I guess that no patient received this treatment? Then it should be N/A.
3. Figure 1: too many significant numbers (4 instead of 3). the percent should be with only one decimal after the point, and point, no coma (Ex. Figure 1 all genotypes, firth percent should be 93.3 instead of 93,33, since only 200 patients: 3 significant numbers).
4. From the data in Table 1, from 200 patients, 196 attained SV4 and 195 SV12. This data is somehow different to the one shown in Figure 1, where more discrepancies seem to exist: for example, all genotypes 93.3% vs 99%, and then EBR/GZR 99% vs 90, which is in contradiction with the first two columns.
5. In addition, several other studies have analyzed this correlation, with a higher number of samples, which is somehow low in this study. For ex. J Med Virol. 2024 May;96(5):e29675. doi: 10.1002/jmv.29675.
6. There are also great disparities among the number of patients in each antiviral treatment, which may have introduced some bias in the results. This is not discussed. The authors should try to increase at least the number of patients on EBV/GZR for a more proper study.
Author Response
Responses to Reviewer 2
Dear Reviewer 2,
The authors would like to thank for the detailed review of the manuscript. They hope that they have adequately addressed your requests.
We would like to inform the reviewers that the revised version of the manuscript includes an additional 6 patients (3 with GT1b, 2 with GT1a, and 1 with GT3) due to a subsequent evaluation of the data. The final number of included participants was 206. All statistical data have been appropriately revised. We hope that this addition further enhances the informativeness of our study.
Comments 1. Materials and Methods, page 2, lines 93-98. Some sentences are confusing throughout the manuscript, particularly the using of sensitivity and specificity for SV4 and SV12. For example: ¨ Negative predictive value (NPV) was characterized as the percentage of patients lacking SVR12 among those who did not attain SVR4¨.
Response 1. The sentence has been rephrased to be clearer: “The negative predictive value (NPV) was characterized as the percentage of patients who did not achieve SVR12 among those who failed to reach SVR4. The negative predictive value (NPV) was characterized as the percentage of patients who did not achieve SVR12 among those who failed to reach SVR4. Sensitivity is characterized as the proportion of patients with SVR4 among those who ultimately achieved SVR12. Specificity is defined as the proportion of patients without SVR4 among those who did not achieve SVR12.” (page 2, lines 102-106).
Comments 2. Figure 1: Genotype 1A. The number 0 is not correct. I guess that no patient received this treatment? Then it should be N/A.
Response 2. We agree, and we have corrected this unintentional error. In the revised version, the corrected Figure 1 is included. All places where necessary are marked with "N/A.
Comments 3. Figure 1: too many significant numbers (4 instead of 3). the percent should be with only one decimal after the point, and point, no coma (Ex. Figure 1 all genotypes, firth percent should be 93.3 instead of 93,33, since only 200 patients: 3 significant numbers).
Comments 4. From the data in Table 1, from 200 patients, 196 attained SV4 and 195 SV12. This data is somehow different to the one shown in Figure 1, where more discrepancies seem to exist: for example, all genotypes 93.3% vs 99%, and then EBR/GZR 99% vs 90, which is in contradiction with the first two columns.
Response 3. We have recognized the errors identified by the authors. In response, a comprehensive statistical review was performed during the evaluation of the revised manuscript, and all errors have been corrected. The updated version now features a corrected Figure 1, which has been carefully aligned with the data in Table 1.
Comments 5. In addition, several other studies have analyzed this correlation, with a higher number of samples, which is somehow low in this study. For ex. J Med Virol. 2024 May;96(5):e29675. doi: 10.1002/jmv.29675.
Response 5. Thank you for pointing out the latest reference; the authors have included it in the manuscript (page 10, lines 312-317). The research from the reference is the result of work by colleagues from a much more populous country, and the sample sizes of both this study and the manuscript you reviewed are proportional when compared to the population size. We firmly believe that it is important to have such results from diverse settings, as this is crucial for the broader acceptance of the research findings. The authors are particularly pleased that their conclusions align with the results from the mentioned reference.
Comments 6. There are also great disparities among the number of patients in each antiviral treatment, which may have introduced some bias in the results. This is not discussed. The authors should try to increase at least the number of patients on EBV/GZR for a more proper study.
Response 6. This can be explained by the inclusion of participants. Participation in the study was entirely voluntary, and only motivated patients returned four weeks after completing therapy for blood sampling for biochemical analyses and SVR4. Although over 500 patients were treated during the study period, only those who chose to participate were included, and their results were analyzed. In addition, GT 1a was the most common, while on the other hand, EBR/GZR has been available significantly longer than pangenotypic drugs, so a larger number of patients with GT 1b have already been treated. The authors are fully aware that this is a weakness of the manuscript, but they believe that even these non-affirmative results, with appropriate explanation, will serve as an encouragement for colleagues to pursue further research. We are very grateful for your effort, as it encouraged us to put in additional work that improved the manuscript. Based on your review, we sought statistical expertise, and it has been concluded that there was an error in the work, which has now been corrected. Additionally, six more participants have been added to the study: 3 with GT1b, 2 with GT1a, and 1 with GT3. All of this has improved the results and confirmed our findings. Therefore, we hope that the revised version of the manuscript will fulfill the criteria for publishing.
The following changes have been made in the text (Discussion, page 10, lines 312-317.): The authors recognize this as a limitation of the study; however, they contend that even these findings, when properly contextualized, can provide valuable insights and encour-age further research in the field. It is also important to consider that the treatment of pa-tients was confined to university centers, due to funding constraints. While this is not an isolated occurrence, it significantly contrasts with practices in some other countries.
Reviewer 3 Report
Comments and Suggestions for Authors
Critical Revision of the Paper
Strengths
-
Relevance and Impact: The study addresses an important clinical question by comparing the predictive value of SVR4 and SVR12 in chronic hepatitis C treatment. This is pertinent given the global effort to eliminate HCV as a public health issue by 2030.
-
Study Design: The prospective design and the inclusion of a significant number of participants (200 patients) add to the robustness of the study. Conducting the study across two university clinics enhances the generalizability of the findings.
-
Comprehensive Data Collection: The study includes detailed demographic data, comorbidities, liver fibrosis stages, HCV genotypes, and treatment regimens. This comprehensive dataset allows for a thorough analysis of factors influencing treatment outcomes.
-
Statistical Analysis: The use of both descriptive and inferential statistical methods, including Cox proportional hazards models, provides a solid framework for identifying predictors of SVR4 and SVR12.
-
Clinical Implications: The findings suggest that SVR4 could potentially replace SVR12 in certain patient populations, which could simplify and expedite HCV treatment protocols, reducing the risk of patient loss to follow-up.
Weaknesses
-
Sample Size and Diversity: While the sample size is adequate, the study lacks diversity in terms of geographical and socio-economic representation, which may limit the applicability of the findings to broader populations.
-
Limited Genotype Specificity: The study notes a lower specificity for genotype 1b treated with EBR/GZR therapy, but does not explore the reasons for this discrepancy in depth. A more detailed analysis of this subgroup is needed.
-
Follow-up Duration: The study follows patients only up to 12 weeks post-treatment. Longer follow-up would be beneficial to assess the durability of SVR4 as a predictive marker for SVR12 and beyond.
-
Potential Confounders: Although the study adjusts for several confounders, there may be other unmeasured variables influencing the outcomes, such as adherence to medication and socio-economic factors.
-
Generalizability: The study was conducted in a controlled clinical environment at two university clinics, which may not reflect real-world settings where patient management and follow-up might differ.
-
Ethical Considerations: The study mentions ethical approval and informed consent but does not detail how patient confidentiality was maintained or any potential conflicts of interest.
Proposed Final Decision
Based on the strengths and weaknesses identified, the study provides valuable insights into the potential use of SVR4 as an early predictor for SVR12 in chronic hepatitis C treatment. However, before incorporating SVR4 into clinical practice guidelines, further research is needed to validate these findings in larger and more diverse populations, and over longer follow-up periods.
Therefore, the paper should be accepted with minor revisions. The authors should address the following points:
-
Expand on Genotype 1b Findings: Provide a more detailed analysis of the lower specificity observed for genotype 1b treated with EBR/GZR therapy.
-
Discuss Long-term Outcomes: Include a discussion on the need for longer follow-up studies to confirm the durability of SVR4 as a predictor.
-
Enhance Generalizability: Discuss the potential limitations related to the study's controlled clinical environment and propose how these findings could be validated in more diverse, real-world settings.
-
Ethical Clarifications: Provide more detailed information on how patient confidentiality was maintained and disclose any potential conflicts of interest.
By addressing these revisions, the paper will offer a more comprehensive and reliable contribution to the field of HCV treatment optimization.
Comments on the Quality of English LanguageMinor changes in English language must be done.
Author Response
Responses to Reviewer 3
First and foremost, the authors would like to thank the reviewer for their critical approach and the effort invested in the review. With great appreciation, we have received your positive feedback, and in the section addressing the critiques, we are also providing responses to them.
We would like to inform the reviewers that the revised version of the manuscript includes an additional 6 patients (3 with GT1b, 2 with GT1a, and 1 with GT3) due to a subsequent evaluation of the data. The final number of included participants is 206. All statistical data have been appropriately revised. We hope that this addition further enhances the informativeness of our study.
Critical Revision of the Paper
Strengths
- Relevance and Impact: The study addresses an important clinical question by comparing the predictive value of SVR4 and SVR12 in chronic hepatitis C treatment. This is pertinent given the global effort to eliminate HCV as a public health issue by 2030.
- Study Design: The prospective design and the inclusion of a significant number of participants (200 patients) add to the robustness of the study. Conducting the study across two university clinics enhances the generalizability of the findings.
- Comprehensive Data Collection: The study includes detailed demographic data, comorbidities, liver fibrosis stages, HCV genotypes, and treatment regimens. This comprehensive dataset allows for a thorough analysis of factors influencing treatment outcomes.
- Statistical Analysis: The use of both descriptive and inferential statistical methods, including Cox proportional hazards models, provides a solid framework for identifying predictors of SVR4 and SVR12.
- Clinical Implications: The findings suggest that SVR4 could potentially replace SVR12 in certain patient populations, which could simplify and expedite HCV treatment protocols, reducing the risk of patient loss to follow-up.
Weaknesses
- Sample Size and Diversity: While the sample size is adequate, the study lacks diversity in terms of geographical and socio-economic representation, which may limit the applicability of the findings to broader populations.
Response 1. The authors fully agree with the stated facts. We believe that more studies like this one, conducted across different geographical areas (one of the first studies comes from New Zealand), could eventually allow for a meta-analysis and more comprehensive conclusions. The authors are from a country classified as a middle to upper-middle-income country; therefore, they consider the data valuable for countries with a similar or richer background. Again, the data are comparable with the study provided by authors from more economically developed nation (New Zealand is classified as a high-income country). Finally, considering that the most common HCV genotypes in the analyzed population are 1 and 3, which is also the case globally, the study's results have the potential for application in other settings.
- Limited Genotype Specificity: The study notes a lower specificity for genotype 1b treated with EBR/GZR therapy, but does not explore the reasons for this discrepancy in depth. A more detailed analysis of this subgroup is needed.
Response 2. The authors absolutely agree with this comment and have highlighted it as a limitation of the study in the discussion. Therefore, they emphasized that it would be ideal to perform SVR4 in patients treated with pangenotypic drugs using a "test and treat" approach. According to unpublished data from the authors, the cure rate with this drug is 98.5% in a much larger sample (the drug has been available since 2018, well before pangenotypic drugs). The authors can only speculate on the reasons for the lower cure rate observed in this study - possible factors include a small sample size, poor compliance, or reinfection during treatment.
- Follow-up Duration: The study follows patients only up to 12 weeks post-treatment. Longer follow-up would be beneficial to assess the durability of SVR4 as a predictive marker for SVR12 and beyond.
Resposnce 3. Although there are studies that have analyzed SVR24, since SVR12 has been established as an endpoint in the treatment process in DAAs era, the authors considered that monitoring SVR4 durability beyond 12 weeks was not necessary according to EASL and AASLD. They do believe it is important, but as highlighted in the manuscript, primarily for patients with F3 and F4 in terms of HCC screening.
- Potential Confounders: Although the study adjusts for several confounders, there may be other unmeasured variables influencing the outcomes, such as adherence to medication and socio-economic factors.
Response 4. Reviewer 3 has highlighted important factors that influence treatment outcomes. The authors share this view and consider all the listed factors (adherence to medication and socio-economic factors) important for treatment outcomes. Given the goal of simplifying the management of patients with HCV infection to achieve global elimination, the authors believe that a model allowing this safety should be found. In this context, they do not recommend universal application of SVR4 based on their findings, but rather suggest it for younger individuals without significant fibrosis. With this approach, patient education importance of adequate compliance is both exceptionally important and indispensable. Additionally, DAAs are very convenient in terms of the number of tablets and daily doses.
- Generalizability: The study was conducted in a controlled clinical environment at two university clinics, which may not reflect real-world settings where patient management and follow-up might differ.
Response 5. Reviewer 3's comment reflects one of the key issues in the HCV cascade of care - the possibility of diagnosis and treatment being limited to university centers only. Decentralization of care is one of the critical aspects that needs to be worked on and is a prerequisite for achieving WHO goals. Treating in university centers does reflect reality in many different clinical settings across the globe (at least in middle income countries), which is why the authors dare to present these results as real-world data. Possible decentralization requires engaging many stakeholders, hence the authors emphasized that this is their small effort to simplify the HCV cascade of care as much as possible from the perspective of clinical practitioners.
To address any potential confusion regarding the possible bias of treatment at a university center and to highlight that this is real-world experience, we have made the following changes to the text.
- Ethical Considerations: The study mentions ethical approval and informed consent but does not detail how patient confidentiality was maintained or any potential conflicts of interest.
Response 6. Patients were thoroughly informed about the protocol and treatment plan. Participation in the study was a matter of the patient's own decision. Only motivated patients returned four weeks after completing therapy for blood sampling for biochemical analyses and SVR4. During the same period when the study was conducted, over 500 patients were treated, but only those included in the study chose to participate, and their results were analyzed. Throughout the study, patients' personal data were never exposed to anyone other than the doctors working with the patients. Personal data were not used during the analysis or presentation of results. A relationship based on professionalism and trust existed between the doctors involved in the study and the treated patients, and this relationship was not compromised during, before, or after the study.
One of the main obstacles was the fact that the authors had no funding, either from their institution or from other sources. The authors personally finance the printing of their manuscripts, which is a significant issue in their professional work. Given that there is no funding or financial support for producing this manuscript from any source other than the personal investment of the authors themselves, and that their work is the result of enthusiasm and a desire to share their experience, they are free to conclude that there is no conflict of interest.
Proposed Final Decision
Based on the strengths and weaknesses identified, the study provides valuable insights into the potential use of SVR4 as an early predictor for SVR12 in chronic hepatitis C treatment. However, before incorporating SVR4 into clinical practice guidelines, further research is needed to validate these findings in larger and more diverse populations, and over longer follow-up periods.
Therefore, the paper should be accepted with minor revisions. The authors should address the following points:
- Expand on Genotype 1b Findings: Provide a more detailed analysis of the lower specificity observed for genotype 1b treated with EBR/GZR therapy.
In accordance with the above, we have applied your advice.
Response 1. First and foremost, it is a small sample. The results are influenced by the limited number of patients with G1b treated with EBR/GZR. The same applies to the established low sensitivity and specificity in patients with GT1b treated with the SOF/VEL combination
- Discuss Long-term Outcomes: Include a discussion on the need for longer follow-up studies to confirm the durability of SVR4 as a predictor.
Response 2. We agree that this could potentially be important for academic reasons; however, this study focuses on clinical needs.
Although there are studies that have analyzed SVR24, since SVR12 has been established as an endpoint in the treatment process in DAAs era, the authors considered that monitoring SVR4 durability beyond 12 weeks was not necessary according to EASL and AASLD. They do believe it is important, but as highlighted in the manuscript, primarily for patients with F3 and F4 in terms of HCC screening.
- Enhance Generalizability: Discuss the potential limitations related to the study's controlled clinical environment and propose how these findings could be validated in more diverse, real-world settings.
Response 3. The following changes have been made in the text:
(Discussion, page 10, lines 313-317): The authors recognize this as a limitation of the study; however, they contend that even these findings, when properly contextualized, can provide valuable insights and encourage further research in the field. It is also important to consider that the treatment of patients was confined to university centers, due to funding constraints. While this is not an isolated occurrence, it significantly contrasts with practices in some other countries.
These findings correspond to the only clinical practice in the Republic of Serbia. Where HCV is treated exclusively in university centers. The same practice has been established in other countries in this part of Europe.
- Ethical Clarifications: Provide more detailed information on how patient confidentiality was maintained and disclose any potential conflicts of interest.
Response 4. The following changes have been made in the text:
(Materials and methods, page 3, lines 110-112): Participation in the study was voluntary. It included patients who were motivated to return for an additional examination and blood sampling four weeks after completing the therapy.
(Discussion, page 10, lines 312-314): The authors recognize this as a limitation of the study; however, they contend that even these findings, when properly contextualized, can provide valuable insights and encour-age further research in the field.
By addressing these revisions, the paper will offer a more comprehensive and reliable contribution to the field of HCV treatment optimization.
Round 2
Reviewer 3 Report
Comments and Suggestions for Authors
Strengths:
-
Timely and Relevant Research: The study addresses a crucial aspect of hepatitis C management, focusing on the potential to shorten the monitoring period for sustained virological response (SVR) from 12 weeks to 4 weeks. This is particularly relevant given the global efforts to eliminate HCV by 2030 as per WHO guidelines.
-
Robust Methodology: The study was conducted in two university infectious disease clinics, which adds to its credibility. The inclusion of 200 patients, the use of standardized diagnostic tools (e.g., liver stiffness measurement, FibroScan®), and the application of established treatment protocols align with current best practices.
-
Comprehensive Data Analysis: The use of both descriptive and inferential statistical methods, including the Cox proportional hazards model, demonstrates a thorough approach to analyzing the factors associated with achieving SVR4 and SVR12. The study’s statistical rigor enhances the reliability of the results.
-
Clinical Implications: The identification of younger age and less advanced liver fibrosis as independent predictors for achieving SVR at both 4 and 12 weeks offers practical insights for clinicians. The suggestion that a shorter monitoring period could reduce loss to follow-up is a valuable contribution, particularly in the context of managing HCV in resource-limited settings.
-
Alignment with Global Health Goals: The study’s findings align with the WHO’s "test and treat" approach, which aims to simplify and enhance HCV treatment protocols. The potential to integrate these findings into official guidelines could significantly impact global HCV management.
Deficiencies:
-
Limited Generalizability: The study was conducted in only two clinics within Serbia, which may limit the generalizability of the findings to other regions with different healthcare systems, patient demographics, and treatment accessibility.
-
Small Sample Size for Subgroup Analysis: While the overall sample size of 200 patients is reasonable, the study could benefit from a larger sample size for subgroup analyses. For example, the low number of patients with genotypes 2 and 4 or those treated with specific antiviral therapies (e.g., elbasvir/grazoprevir) may reduce the statistical power and robustness of the findings in these subgroups.
-
Potential Bias in Patient Selection: The study includes only patients who were motivated to return for follow-up at 4 weeks. This self-selection bias could result in an overestimation of SVR rates and may not accurately represent the broader population of HCV patients, particularly those who are less compliant or have socio-economic barriers to healthcare access.
-
Lack of Long-term Follow-up: The study focuses on SVR at 4 and 12 weeks post-treatment but does not provide data on long-term outcomes, such as SVR at 24 weeks (SVR24), recurrence rates, or the progression of liver disease in the months or years following treatment.
-
Inconsistent Specificity Results: The study reports a maximum sensitivity of 100% for achieving SVR12 after SVR4 across all genotypes but notes variability in specificity, particularly for patients with genotype 1b treated with elbasvir/grazoprevir, where specificity was only 66.7%. This inconsistency suggests that the 4-week monitoring period may not be universally applicable across all patient groups and treatment protocols.
-
Need for Further Validation: The study itself acknowledges the necessity for further research to validate these findings. This is crucial before any changes can be confidently recommended for clinical practice or incorporated into official treatment guidelines.
Final Decision:
The paper offers significant insights into the potential for shortening the post-treatment monitoring period for HCV patients, which could have practical implications for improving patient outcomes and reducing healthcare costs. However, the study's limitations, particularly in terms of generalizability, sample size, and long-term follow-up, indicate that while the findings are promising, they are not yet sufficient to warrant changes to current clinical practice.
Comments on the Quality of English LanguageMInor changes in English language must be done.